# Experiences of Student Nurses Regarding Experiential Learning in Specialized Psychiatric Hospitals in South Africa, a Qualitative Study

**DOI:** 10.3390/healthcare11081151

**Published:** 2023-04-17

**Authors:** Thingahangwi Cecilia Masutha, Mary Maluleke, Ndidzulafhi Selina Raliphaswa, Mphedziseni Esther Rangwaneni

**Affiliations:** Department of Advanced Nursing, The University of Venda, Private Bag X5050, Thohoyandou 0950, South Africa

**Keywords:** personal experiences, experiential learning, student nurse, specialized psychiatric hospitals

## Abstract

Background: Experiential learning in specialized psychiatric hospitals is as essential as other learning in any discipline in nursing education as it allows student nurses to correlate theory to practice. Experiential learning in the mental health environment has been identified as a significant factor in promoting a more favorable attitude among student nurses toward mental health nursing. Aim: The study explored student nurses’ personal experiences regarding experiential learning in specialized psychiatric hospitals. Methods: A qualitative approach using explorative, descriptive, and contextual designs was used, and 51 student nurses were purposively sampled. Data were collected through six focus group interviews and analyzed using a thematic approach. Measures to ensure trustworthiness were also enhanced. Ethical considerations were adhered to throughout the study. Results: One theme and four sub-themes on the experiences of student nurses regarding experiential learning in specialized psychiatric hospitals were revealed, namely: theme: personal factors; sub-themes: fear of mental healthcare users, anxiety about clinical assessment, lack of interest in psychiatric nursing science, and stress due to social problems. Conclusions: Based on the findings, student nurses experience many things during experiential learning, including personal factors. A further qualitative study should be conducted on strategies to support student nurses during experiential learning in the specialized psychiatric hospitals of Limpopo Province.

## 1. Introduction

Experience-based education in psychiatric hospitals has been identified as a key element in fostering student nurses’ positive attitudes toward mental health nursing [1]. This aligns with the South African Nursing Council Nursing Act [2], which shows that the clinical skills and competencies of student nurses in psychiatric hospitals are a necessary and mandatory requirement. Students are prepared to become safe and independent professional nurses after graduation through exposure to experiential learning in psychiatric hospitals and the development of clinical skills. The Nursing Act [2] further stipulates that student nurses must acquire experiential learning skills and competencies before registration as professional nurses. Therefore, it is imperative to expose student nurses to experiential learning opportunities according to the professional body’s requirements before graduation. Similarly, nursing education and training standards under the South African Nursing Council (SANC) Nursing Act [2] state that student nurses are expected to acquire the knowledge, skills, and behaviors that meet SANC standards to deal with present and future challenges and improve health and welfare.

Another study revealed that the root causes of absenteeism among nursing students were indiscipline, dependency, a great deal of expectation from teachers and parents among students, a lack of cooperation and coordination between parents and the institution, misunderstanding between parents, a lack of knowledge among parents in handling adolescent children, and a lack of timely guidance by the teachers in the clinical and theory classes [3].

The Irish Nursing Council [3] requires that nursing students attend all of their clinical rotations each year. Finland also accepts 95% attendance to fulfill course requirements [4]. Similarly, the Nigerian University of Abadan requires nursing students to acquire 75% clinical attendance [5]. On the other hand, the South African Nursing Council [2] states that student nurses are expected to accumulate 80% of clinical hours before registration as professional nurses. According to the findings of the study conducted in Iran, nursing students reported that their biggest issue was identity confusion as a result of the inadequate clinical setting. Nursing students realized that the clinical training environment lacked the necessary efficiency when they were there. This ineffective educational environment was made up of ineffective teachers, unfavorable lesson ideas, and an improper clinical setting [6].

However, upon the end of experiential learning, most student nurses lack the expected clinical hours in higher education, especially in nursing education institutions [7]. According to another study [8], being absent from the classroom, where students are expected to be present, is extremely concerning [8]. It has been observed that student nurses at Limpopo College of Nursing had a higher rate of repeating a year of study due to a shortage of psychiatric clinical hours between 2018 and 2021, which indicates that level IV had a higher rate in 2018 and 2020 as compared to level III. In contrast, level III had a higher rate in 2019 and 2021 than level IV [9]. Despite all of the government’s efforts, nursing schools around the world are becoming concerned about the lack of psychiatric clinical hours [8]. An Bord Altranais found that 97.3% of nursing students work few hours in this clinical area [4]. 

A shortage of psychiatric clinical hours has a negative impact on student nurses and nursing practice. Another study shows that a lack of hours results in missed opportunities to develop the skills and knowledge necessary for successful professional practice. Students’ academic and professional advancement is hampered when they miss lectures since they lose educational learning information that may be crucial for the exam and professional practice [10]. Chukwu et al. reported that if students do not get enough time to experience learning, they will not be able to achieve their personal and professional goals [11]. Furthermore, it was revealed that a lack of mental clinical hours affects students negatively and lengthens their study time. Additionally, it conveys a lack of enthusiasm for and interest in academic pursuits. On the other hand, a study revealed that illness, a lengthy distance to school, breastfeeding a baby, a lack of interest in lectures and clinical experience, a lack of transportation, and non-payment of school fees are contributing factors to absenteeism among nursing students [12]. Similarly, another study shows that the shortage of clinical hours has an adverse effect on the quality of nursing care that students must render [6]. According to the South African Nursing Council, a student who has obtained less than 80% of experiential learning hours is not allowed to sit in on the final examinations of a subject [2]. When student nurses are absent from experiential learning, they are short of stipulated clinical hours, and as a result, the students must repeat a level of study for a year. Consequently, fewer students graduate at the end of four years, leading to a shortage of professional nurses in the country. A study was conducted in Iran about the challenges faced by nursing students in the clinical learning environment [13]. On the other hand, Egypt conducted a study of nursing students’ experiences in psychiatric clinical practice [14]. Another study was conducted about clinical placement experiences by undergraduate nursing students in selected teaching hospitals in Ghana [15]. Similarly, South Africa studied nursing students’ experiences regarding clinical placement and support in primary healthcare clinics [16].

The aforementioned research focused on the student nurses’ experiences in a clinical setting generally, in clinics rather than psychiatric hospitals, in nations other than South Africa, and in other regions except for Limpopo Province. So, the study examined student nurses’ personal experiences with experiential learning at South African specialty mental facilities.

## 2. Materials and Methods

### 2.1. Study Design and Setting

A qualitative approach was employed using explorative, descriptive, and contextual designs [17]. A qualitative approach was adopted because the study sought to express students’ experiences regarding experiential learning in specialized psychiatric hospitals in South Africa. The study was conducted at the three selected specialized psychiatric hospitals (Evuxakeni, Hayani, and Thabamoopo), which were campuses of the Limpopo College of Nursing and provided student nurses with opportunities for experiential learning. The Evuxakeni specialized psychiatric hospital is situated on the main road, Giyani, about 5.3 km outside the town of the Greater Giyani sub-district, which is made up of rural areas with a population of different cultures. Sotho, Tsonga, and Venda-speaking people are the dominant groups. The Hayani specialized psychiatric hospital is situated in the Sibasa rural area on the main road, next to Thohoyandou city. Venda-speaking people dominate it. The Thabamoopo psychiatric hospital is situated in the Lepelle Nkumpi local municipality in the rural area of Lebowakgomo, in the south-east, just 50 km from Polokwane. The inhabitants speak Sepedi as their first language. Student nurses travel by bus from the Giyani Campus to the Evuxakeni Hospital, 5.3 km away. From the Thohoyandou Campus to the Hayani Hospital, it is 13.6 km, and from the Sovenga Campus to the Thabamoopo Hospital, it is 49.5 km. The University of Limpopo and the University of Venda also use the Thabamoopo and Hayani psychiatric hospitals to place student nurses for their practical studies. 

### 2.2. Study Population and Sample

A total of 51 student nurses who had been absent from the specialized psychiatric hospitals before were purposively sampled from the three hospitals, as indicated by the attendance registers of the selected hospitals, whose details are complete in the current year of study. A total of 30 participants were at level IV, while 21 were at level III. Sampling occurred in two stages, namely, sampling of hospitals and sampling of participants using the following criteria: only hospitals that receive level III and IV students from the Limpopo College of Nursing. Participants who were in levels III and IV and only those who had been absent from the three selected hospitals were sampled. The participants were visited in their respective hospitals to recruit them a day before the interviews. The purpose and nature of the study were explained to the potential participants, as outlined in the participant’s information sheet. 

All participants’ names and telephone numbers were listed, and they were contacted telephonically before interviews, visited at the hospitals for recruitment, and the information regarding the study was explained briefly. The participants and the researcher agreed upon the date, time, and venue. The researcher’s contact numbers were given to the participants in case there were changes regarding the arrangements. The participants were informed that they were not forced to participate in the study and that if they did participate, they had the right to withdraw at any interview stage. It was also explained that an audio recorder would be used to record the conversation, and they were shown a stop button to stop the recording at any time if they wished to do so.

### 2.3. Data Collection

Focus groups of semi-structured interviews were used to collect data, and the anticipated number of student nurses in each focus group from each hospital was from eight to ten. Focus groups are semi-structured conversations among groups of four to twelve people with the goal of examining a certain set of issues [18]. Ten level IV student nurses from Hospital 1, eight from Hospital 2, and twelve from Hospital 3 were interviewed in three focus groups, making thirty participants who volunteered and consented to participate. Seven level III students were interviewed from Hospital 1, six from Hospital 2, and eight from Hospital 3, bringing the total to twenty-one participants who volunteered and consented to participate. Thus, the overall number of student nurses interviewed was fifty-one. Then the interviews were transcribed. Data were collected in two months. It took one focus group per day as the institutions are far apart, and each focus group lasted from 45 min to an hour. An audio tape was used to record the interviews. Data saturation was reached in the fifth focus group, but the researcher continued to the sixth focus group to ensure that no new information was left out.

### 2.4. Data Analysis

A tool developed by Tesch guided data analysis. Tesch provides eight steps, outlined in [18], that should be considered when analyzing qualitative data using a thematic approach. A coding scheme was used to identify the themes evident from the data. The first author deduced the information from the collected data. A theme was allocated to a specific sub-theme that captured the same idea. The panel of experts of authors scrutinized the deduced information and themes to increase validity. The authors then discussed the findings and reached a consensus. After reaching a consensus regarding the theme and sub-themes, they were then categorized into a theme and sub-themes that emerged from the data, namely: a theme of personal factors; sub-themes: fear of mental healthcare users, anxiety about clinical assessment, lack of interest in psychiatric nursing science, and stress due to social problems.

### 2.5. Ethical Considerations

Permission was sought from the following stakeholders: the University Higher Degree Committee, the Research Ethics Committee of the University of Venda (SHS/20/PDC/04/1305), the Limpopo Province Department of Health, the CEOs of the three selected hospitals, and participants who signed consent forms before participating in the study. The environment where the interviews were conducted was safe for participants since it was in the hospitals where they were allotted. Therefore, no harm was done to the participants. The participants were assured of confidentiality. An agreement with them was respected, including punctuality of the time agreed upon. Names were not used; only the participants’ numbers were used for anonymity. The COVID-19 protocols were also adhered to.

## 3. Results

Level III and IV student nurses from selected specialized psychiatric hospitals were sampled. The sample comprised thirty level IV student nurses and twenty-one level III student nurses from the three selected hospitals. The sample size of this study was determined by data saturation, which was reached after interviewing the sixth focus group. Focus groups, hospitals, and participants were numbered: focus groups 1–6, hospitals 1–3, and participants 1–10. 

During data analysis, theme 1 emerged when participants shared their personal experiences influencing student nurses’ experiential learning. This theme identified four sub-themes: fear of mental healthcare users, anxiety about clinical assessment done in the clinical areas, lack of interest in psychiatric nursing science, and stress due to social problems. Table 1 shows the theme and sub-themes identified.

Fear of mental healthcare users.

Most student nurses indicated they were uncomfortable in the wards, leading to being absent from psychiatric hospitals because of their fear of mental healthcare users, particularly those students exposed to psychiatric wards for the first time. The following quotations indicate student nurses’ experiences of fear of mental healthcare users:

*“…Ok, err, I wanted to mention safety concerns as one of the contributory factors. Like some students fear mental healthcare users, especially because some of them have got a terrifying history. Ok, what do I mean by saying terrifying history I mean, aah, the thing is sometimes, ok, obviously, we have access to their files, so sometimes we find out that the person has got a history of maybe rape times 4 and there is a lady who is supposed to be working around those people every day, so that can result to that person being afraid of such people. Another example I can give is an example of someone who is admitted and has a history of murder x 5, so obviously I won’t feel safe around those people…”*.Participant no 8—Focus group no 1

Another participant said:

*“…No, I did not want to say something, but then I can add on what participant no 8 was saying. So sometimes those patients tend to call the female students as if they are in an intimate relationship; you will find they are calling them “my lover” and so what, so I feel like some of the female students have a fear of being at work because they feel like their safety is being violated…”*.Participant no 9—Focus group no 1

Another participant mentioned that:

*“…And the behavior of the mental health care user towards the students. Is it indeed that, like those mental health care users, they spend most of their time in the ward, so they know that these are students’ nurses and these are staff, so you may find that the way they treat us is unacceptable to such an extent that err we become reluctant to go to work…”*.Participant no 1—Focus group no 2

Anxiety about clinical assessments performed in the clinical areas.

Some participants mentioned that they were anxious about clinical assessments performed by nurse educators in the clinical areas; they felt they were not ready to be assessed. The following quotations indicate student nurses’ experiences of anxiety about clinical assessments performed in the clinical areas:

One participant said:

*“…Ok, it is about having anxiety about the assessment that should be done by the student being assessed by the nurse educators. You are not ready then, but the nurse educators who should assess you will say I told you I would come today…”*.Participant no 7—Focus group no 5

Another participant said:

*“...Yah but she said most of the things I wanted to say but we are expected to study in a month everything and we write like every week we are writing and on the other hand you are being assessed, so it is just too much workload so when we get time you absent yourself and you come back you study to catch up, yah…”*.Participant 6—Focus group 3

Another participant said:

*“…You will find that your lecturer is coming for assessment and then you speak with the mental health care user and when the assessor comes, he changes the story he doesn’t want to do the assessment anymore and then when you tell the assessor she doesn’t understand and then yah…”*.Participant no 3—Focus group no 5

Lack of interest in psychiatric nursing science

Other students indicated that student nurses show a negative attitude towards PNS; they are not interested in specific subjects while training. The following quotations indicate student nurses’ experiences of a lack of interest in psychiatric nursing science:

One participant said:

*“…Yah, I just want to add some information on what she has just said; I think if more people have to choose on their own, unlike this one, if you have to become a professional nurse, you must have a psychiatric bar with you, I think it becomes simpler or more interesting because it is you willing to do that course on your own…”*.Participant no 3—Focus group no 1

Another participant said:

*“…Err, just to add, because definitely if it is my choice actually to participate or enter that psychiatric field, it means that I will be more interested to learn whatever that it is eh to learn in that field, which means I will be competent enough in that field to become err a fire psychiatric nurse…”*.Participant no 7—Focus group no 1

Another participant mentioned that:

*“…Another thing is attitudes of us, students towards psychiatry because you find that is indeed that at the college we are being trained in different faculties, there are different subjects, so you find that a student has attitude towards a certain subject. Then you find that a student doesn’t like psychiatry and told himself that when I complete, I don’t want to work in psychiatry so it makes him loose interest in psychiatry because he doesn’t want to work in it on completion so it also lead to absenting himself from work which is absenteeism…”*.Participant no 8—Focus group no 2

Stress due to social problems

Some participants indicated that student nurses have social issues and undergo stress as social human beings in the community. Participants also indicated that some are married with children and have other family members to take care of. The following quotations indicate student nurses’ experiences of stress due to social problems:

One participant mentioned that:

*“…Also, like things happening to the individual personally, I feel like, at some point, they affect the person’s ability to go to work or not. At least someone going through a stressful situation would like to take a few days off, maybe to recuperate from stress. Maybe even a breakup or you have lost someone whom, people take these things differently, so someone will feel that they don’t want to be around people that is why they absent themselves from work…”*.Participant no 1—Focus group no 3

Another participant said:

*“…I guess it is obvious that the relationship feels like a master and servant because whenever the student is not at work, they are more concerned about the proof that you are to bring more than about what you are going through because, as students, we do have social issues, so they don’t have time to dig into that and try to intervene instead they need, provide proof if you have been to the doctor or provide proof if there has been a funeral at home. They don’t take time to talk to you so that they can know what you have been going through and try to intervene more so that they can find assistance for you; they are just concerned about the proof, that’s it…”*.Participant no 2—Focus group no 4

Another participant mentioned that:

*“…Another thing is that as student nurses, they are also parents from their families, so at their families, they may experience family problems at home. So it may lead to absenteeism. Yes, social problems. You may find that they sometimes fight with their spouses and then they experience communication breakdown then it makes them absent themselves…”*.Participant no 7—Focus group no 1

## 4. Discussion

Most student nurses indicated they were scared of mental healthcare users, who made them stay away from the wards, particularly those students exposed to psychiatric wards for the first time. Similarly, Iran [19] developed strategies to reduce the shortage of hours from the student’s viewpoint, which revealed that negative views of people with mental illnesses as dangerous and harmful were another issue that students brought up during interviews. Some students reported that they were terrified of being physically abused by patients. To transfer teachers’ knowledge, the presence of students is necessary for the classroom and clinical areas. In another study conducted by the researchers [13], they found that another topic raised by students in focus group talks was their perception of those with mental problems as being dangerous and harmful. Other students claimed to be scared of the patients abusing them physically.

Some mentioned that they became anxious about clinical assessments done in the clinical areas; they felt they were not ready to be assessed. This was supported by [15], which indicated that the clinical experiences that undergraduate nursing students had while working in teaching hospitals were varied. Undergraduate nursing students had both good and bad experiences in teaching hospitals. A similar study by [10] shows that reasons for the shortage of hours were identified with assessment factors, such as “I am supposed to demonstrate the procedure,” “when I am supposed to give feedback on evaluation,” and “OSCE day,” so it was concluded that student nurses were short of hours in clinical settings because of fear of assessment. Similarly, the study conducted by [7] found that student nurses expressed that the shortage of hours was due to several factors, including many clinical assignments. Results from a different study by [20] showed a statistically significant correlation between assessment and absenteeism. It is apparent that the prior study has established a link between in-class participation and course grades. The importance of student attendance in ensuring academic success makes it even more important for educators to be well-prepared and aware of how their students study. By examining the relationship between these two characteristics, this study acts as an early intervention for educators to take the appropriate actions. Hence, teachers can lower absenteeism by identifying the issues that their students are having and establishing a productive learning environment.

Another research indicated that student nurses are negative toward psychiatric nursing science; they are not interested in specific subjects while training. Another study by [21] indicated that the convenience of the lecture material, getting ready for exams, a lack of interest, and nurse educators’ teaching methods were the top causes of absenteeism. These findings were also supported by [22], which revealed that illness, a long distance to school, nursing a baby, a lack of interest in lectures and clinical experience, a lack of transportation, and non-payment of school fees are contributing factors to absenteeism among nursing students. Another study by [16] found that the students felt that the patients did not have confidence in them and preferred to have the professional nurse assist them during consultations. They revealed that some patients also showed hostility toward the professional nurses, for instance, when the professional nurse suggested a different course of treatment. A similar study by [23] revealed that student nurses had more negative than positive experiences. The negative experiences included overcrowding by students, student nurses’ negative emotional experiences, challenges of working with professional nurses, and positive experiences of knowledge sharing with and from various healthcare disciplines.

In addition, some participants indicated that student nurses stayed away from the clinical areas because of the social issues and stresses they undergo as social human beings in the community. Participants also indicated that some are married with children and have other family members to take care of. Another study by [24] confirms our findings that several students may have family obligations that place additional demands on their time. Some students have children that require childcare, have aging parents with health problems, or are collecting social grants. In addition, other factors include conflicts at home and family separation, divorce, child self-care, problematic neighborhoods, and maltreatment. Similarly, other findings by [3] indicated that a student revealed that when his father is at home, he makes him get out of bed early and sends him to college, but when he is not, his mother will not make him get out of bed because she does not like that he has decided to study nursing, so he ends up being absent. On the other hand, [4] concludes that students miss class because they are worried about getting paid to study but not to work. If they were paid to work, they would be able to take care of family emergencies like sick children, spouses, or parents as needed. This study was restricted to three hospitals in three districts that may not represent all Limpopo nursing college student nurses.

## 5. Implications

The findings of the study might give direction to college curriculum developers and policymakers for hospitals in the province. The nursing profession is one that directly and significantly affects patient care. It is important that during the training of students, a strict recommendation be established, and students always attain and show key interest in patient management. Absenteeism of any kind will lead to nurses who will not take patient care seriously, and such nurses will be setbacks to the profession.

## 6. Conclusions

The study’s findings contributed to a better understanding of student nurses’ experiences during experiential learning at specialized psychiatric hospitals. The study is important because the hospitals would benefit, and the findings may influence the policymakers of the province for the betterment of patient care. The research may benefit as the study will add information to the body of knowledge. The study is relevant as it was about student nurses’ experiences during experiential learning in psychiatric hospitals but not in other hospitals.

## 7. Recommendations

A further qualitative study should be conducted to develop strategies to mitigate the shortage of hours of experiential learning that student nurses receive at specialized psychiatric hospitals. The three hospitals should provide workshops for healthcare professionals on how to deal with student nurses’ concerns and challenges in the unit. Regular clinical meetings should be held between hospital managers, staff members, lecturers, and students so that clinical areas become aware of the problems that student nurses come across during clinical placement.

## Figures and Tables

**Table 1 healthcare-11-01151-t001:** Results Summary.

Theme	Sub-Themes
Theme 1 Personal experiences influencing student nurses’ experiential learning.	Sub-theme 1 Fear of mental healthcare users.
	Sub-theme 2 Anxiety about the clinical assessment performed.
	Sub-theme 3 Lack of interest in psychiatric nursing science.
	Sub-theme 4 Stress due to social problems.

## Data Availability

The anonymized data are available from the corresponding author upon request.

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
