# Peer review of "Experiences of Student Nurses Regarding Experiential Learning in Specialized Psychiatric Hospitals in South Africa, a Qualitative Study"

_healthcare, 2023, doi:10.3390/healthcare11081151_

Round 1
Reviewer 1 Report
Topic is interesting and the paper is well structured. Would suggest some improvements:
-> Change this paragraph to include the topic of research but not exactly the title: "A study was conducted in Iran by [10] about the Challenges of Nursing Stu-75 dents in the Clinical Learning Environment: A Qualitative Study. On the other hand, 76 Egypt [11] conducted a study of nursing students’ experiences of Psychiatric Clinical Prac-77 tice: A Qualitative Study. Another study was conducted by [12] about clinical placement 78 experiences by undergraduate nursing students in selected teaching hospitals in Ghana. 79 Similarly, South Africa [13] studied nursing students' experiences regarding clinical place-80 ment and support in primary healthcare clinics: Strengthening resilience."
-> Change this sentence to positive: "The above studies’ focus was on the experiences of the student nurses in a clinical 82 environment generally, and in the clinics but not in the psychiatric hospitals, in other 83 countries but not in South Africa, and other provinces but not in Limpopo Province. Hence, the study investigated student nurses' experiences regarding experiential learning 85 in specialized psychiatric hospitals in South Africa."
-> In methodology please define de concept of "Focus group interviews" was this a semistructured individual interview?
The aim of the paper is not quite aligned with the results presented, please state at the beginning which kind of experiences will be analyzed: personal, institutional, emotional?
Reviewer 2 Report
Dear researcher(s), you are addressing an important and meaningful gap. Your paper is well-written and it has some important results, and if you edit your paper it can be much more effective. Here some humble suggestions to improve the paper, I would do the following to strengthen the paper. I have enjoyed reading the paper and am looking forward to seeing the paper published. You could increase the effect of your paper with some more recent studies suggested below or any other studies and not using the suggested ones.

Reviewer 3 Report
The present manuscript explores the experiences of student nurses with experimental learning in specialized psychiatric hospitals.
The paper seems to be well structured and presented. I only suggest carefully re-read the manuscript to correct some English terms and sentences. For instance lines 82-86 seem redundant and unclear.
I also suggest adding a Table that can summarize all the results.
Round 2
Reviewer 2 Report
Dear Researcher (s), the article has been updated by your team to make it clearer and more comprehensive and you are filling a significant and significant gap. This work is well-written and contains some significant findings. I want to congratulate the researchers and the research team on this achievement.